# Investigating the procurement system for understanding seasonal influenza vaccine brand availability in Europe

**Anke L. Stuurman**[1] *, **Caterina Rizzo**[2], **Mendel Haag**[3]

**1** P95 Epidemiology and Pharmacovigilance, Leuven, Belgium, **2** ECDC National Focal Point for Influenza and Other Respiratory Viruses Program, Bambino Gesù Children's Hospital, Rome, Italy, **3** Seqirus, Amsterdam, The Netherlands

* anke.stuurman@p-95.com

## Abstract

### Background

Timely knowledge of which influenza vaccine brands are procured and where is of interest to inform site-selection for brand-specific influenza vaccine effectiveness (VE) studies. Vaccine procurement is a key determinant of brand availability. We therefore sought to understand how the procurement for seasonal influenza vaccine in Europe is organized, how this drives brand availability and how procurement data could enable to determine brand availability pre-season.

### Methods

Structured telephone interviews were conducted with 15 experts in 16 European countries between 2017 and 2019 to collect information on the influenza vaccine procurement systems. Sources of (brand-specific) procurement data were identified and assessed on public accessibility. Vaccine type and brand availability and timelines were determined for the 2019–20 season to understand how procurement systems drive brand availability and diversity.

### Results

Four main types of procurement systems for seasonal influenza vaccination campaigns were identified: national public tenders (Croatia, Denmark, Finland, Ireland, Lithuania, Netherlands, Norway, Scotland, Slovenia), regional public tenders (Italy, Spain, Sweden), direct purchase of vaccines by GPs (England, Wales) or pharmacies (Belgium, France, Germany, Greece) from manufacturers or wholesalers. National public tender outcomes are publicly available and timely; brand availability at clinic level can generally be deduced or narrowed down to two brands. Regional tender outcomes are more difficult to find, known very late or not available. In Italian and Spanish regions tenders may be awarded only a few weeks before the seasonal campaign. No public procurement information is available for countries with direct purchase.

**Data Availability Statement:** All relevant data are within the manuscript.

**Funding:** EU/EFPIA IMI-2 Joint Undertaking (DRIVE, grant n°777363). The funder provided support in the form of salaries for authors AS, CR,

MH, but did not have any additional role in the study design, data collection and analysis, decision to publish, or preparation of the manuscript. The specific roles of these authors are articulated in the 'author contributions' section.

**Competing interests:** MH is an employee of Seqirus. AS is an employee of P95 Epidemiology and Pharmacovigilance. This does not alter our adherence to PLOS ONE policies on sharing data and materials.

## Conclusion

At the country-level, brand diversity is generally lower for countries with national public tenders than for countries with regional public tenders or direct purchase. In only a few countries, procurement data at the brand level is both publicly available and timely. Therefore the usefulness of procurement data for prospective site-selection for brand-specific VE studies is limited.

## Introduction

The European Medicines Agency (EMA) requires marketing authorization holders to annually estimate brand-specific vaccine effectiveness (VE) of their influenza vaccines [1]. VE is a measure of the benefit of the vaccine as determined under real-world circumstances in the routine care setting. To study brand-specific VE, sufficient coverage is needed for the vaccines of interest in the study sites contributing to the data collection.

Timely knowledge on which vaccine brands are available where in any upcoming influenza season is of interest to DRIVE, a public-private partnership funded by the Innovative Medicines Initiative (IMI2) that aims to establish a sufficiently sized network of sites to annually assess brand-specific influenza VE for all influenza vaccines used in the European Union [2]. The prospective identification of locations where specific brands will be available could allow to attain sufficient sample size for each brand in pooled analyses, by prioritizing these locations in the tender selection or by actively approaching sites in these locations.

In Europe, 11 influenza vaccine brands from 6 manufacturers were on the market in 2019–2020. The availability of a specific vaccine brand in a certain location generally depends on multiple factors, such as vaccine license status, recommendations and procurement, i.e. the mechanism(s) at the country, regional and/or clinic level by which vaccines are acquired. Vaccine licensure (whether through a centralized procedure, mutual recognition procedure, decentralized procedure or national authorization [3,4]) is a pre-requisite for marketing, but not indicative of whether a vaccine is available. Specifically, for influenza vaccines in Europe, the procurement is a major determinant of where certain vaccine brands are available in an upcoming influenza season. Vaccine manufacturers have data on sales, however this information is not in the public domain and cannot be readily shared due to competition law that applies to the pharmaceutical industry as it does to any other industry [5]. Limited information exists on the influenza vaccine procurement systems in European countries.

We therefore sought to understand how the different procurement systems for seasonal influenza vaccines in Europe are organized and how procurement data could enable to determine brand availability pre-season. For this purpose, we assessed whether the timing of procurement, and accessibility and content (i.e. brand specifics) of the data source on the procured vaccines in a given country, region or clinic, could support the identification of relevant locations for VE studies.

## Methods

Structured telephone interviews were conducted with experts on influenza vaccine procurement in European Economic Area (EEA) countries. We attempted to identify experts with knowledge on countries with >2 million inhabitants: Austria, Belgium, Croatia, Czech Republic, Denmark, England, Finland, France, Germany, Greece, Ireland, Italy, Lithuania,

Netherlands, Norway, Poland, Portugal, Romania, Scotland, Slovenia, Slovakia, Spain, Sweden and Wales. Hungary was excluded because the influenza vaccine used there is only authorized in Hungary and therefore does not fall under the scope of the EMA guidance. Experts were identified from public institutes (national public health institutes in Slovenia, Norway, Croatia, Netherlands, Denmark, Finland; regional public health institute in Spain; national health insurance fund in Lithuania; University of Oxford) and vaccine manufacturers (Sanofi Pasteur, Seqirus, GlaxoSmithKline). Interviews were conducted between October 2017 and September 2019. Findings for countries with no national tenders were cross-checked with a second expert. Online resources were used to fill information gaps where possible; these were identified using search terms such as influenza, vaccination, and procurement.

The topics covered in the interview were 1) the organization of the influenza vaccination campaigns, 2) the organization of the procurement of influenza vaccines, 3) timing of influenza vaccine procurement, 4) number of vaccine types and brands and historical variability, and 5) public availability of information on procured vaccines (see S1 Appendix for interview guidance used). Where applicable, these aspects were characterized at the national, regional and clinic level. The clinic level refers to the clinical setting where a person is provided the vaccine, either for immediate administration or for administration by another health care provider. The interviews focused on the average situation in these countries. Exceptions may apply in certain situations or across seasons.

The data on vaccine type and brand availability included in this paper reflect the 2019–2020 season. Data on vaccine types and number of brands was collected. Additionally, for countries with national and regional tenders, information on tender issuer, the proportion of brands by vaccine type, volume and sources of public availability of (brand-specific) procurement were collected. Data was obtained through the interviews, through public sources obtained from the experts, and other online public sources. In addition, it was determined whether the publicly available procurement information is informative on brand availability at the level of the clinic.

Vaccines were categorized into conventional trivalent influenza vaccine (TIV), adjuvanted trivalent influenza vaccine (aTIV), high dose trivalent vaccine (TIV-HD), egg-based quadrivalent influenza vaccine (QIVe), cell-based quadrivalent influenza vaccine (QIVc) and live attenuated quadrivalent influenza vaccine (LAIV).

## Results

Of the 24 targeted countries, interviews were completed for 16 countries (Belgium, Croatia, Denmark, England, Finland, France, Germany, Greece, Ireland, Italy, Lithuania, Netherlands, Norway, Slovenia, Spain, Sweden), with 15 experts from national public health institutes (n = 10) and vaccine manufacturing companies (n = 5). For the remaining countries, no suitable expert was identified after repeated attempts, or no response was received. For Scotland and Wales relevant information was found online.

### Organization of the seasonal influenza vaccination campaigns

Vaccination campaigns generally start between the second half of September and the beginning of November.

In most countries, the vast majority of influenza vaccines are administered by general practitioners (GPs) or pediatricians for children. In Denmark, in addition to GPs, dedicated vaccination clinics play an important role in influenza vaccine administration (ca. 25%). In France, eligible patients receive a voucher from the national health insurance, and in Belgium and Greece, patients have to get a prescription from their GP; after this, in those three countries,

they have to obtain the vaccine at a pharmacy before visiting their GP (or another healthcare provider) for the actual vaccine administration, although some pharmacies also administer vaccines. In England and Wales, those routinely recommended for vaccination can obtain their vaccine at their GP, or since 2015–2016 increasingly also at pharmacies [6–8], except for children who attend school who are primarily vaccinated through school-based programs [9]. In Finland, children are primarily vaccinated at well baby clinics.

In some places, such as the United Kingdom, Italy and regions of Spain, type-specific influenza vaccine recommendations for different risk groups are in place [10–13].

## Organization of the procurement of influenza vaccines

Four main types of influenza vaccine procurement systems for seasonal influenza vaccination campaigns were identified: a) national public tenders (Croatia, Denmark, Finland, Ireland, Lithuania, Netherlands, Norway, Scotland, Slovenia), b) regional public tenders (Italy, Spain, Sweden), c) direct purchase of vaccines by GPs from manufacturers (England, Wales), and d) direct purchase of vaccines by pharmacies (Belgium, France, Germany, Greece) from manufacturers or wholesalers (Fig 1). These reflect the primary procurement systems that are in place.

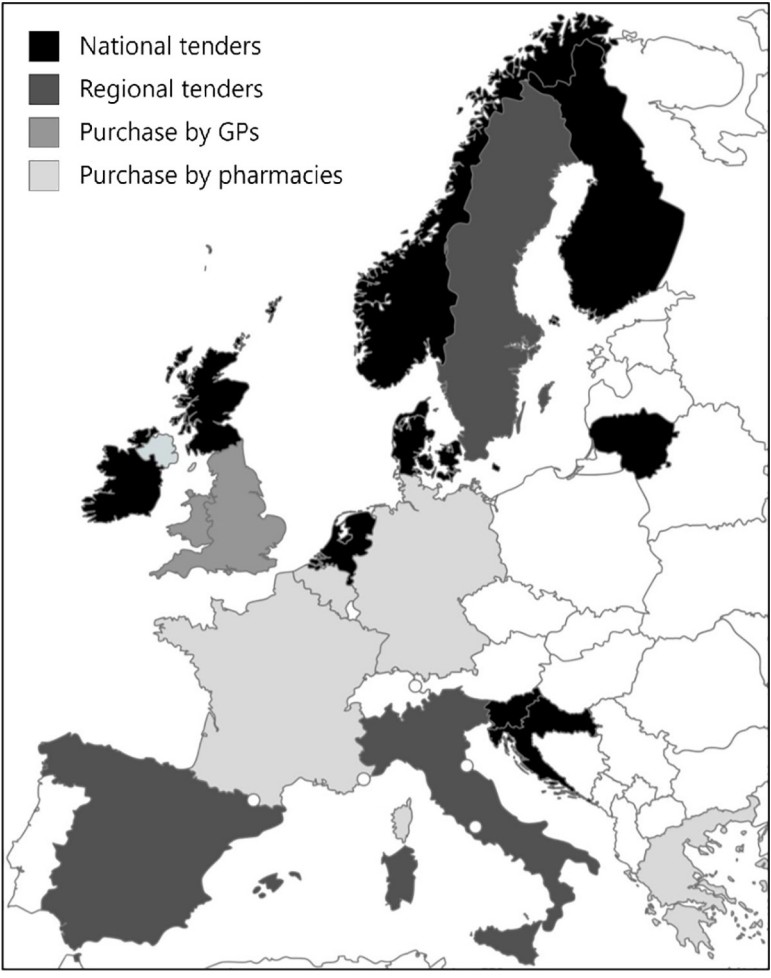

**Fig 1. Four main procurement systems for seasonal influenza vaccines in 18 European countries.** Republished from mapchart.net under a CC BY license, with permission from Minas Giannekas, original copyright 2021.

National public tenders are issued by the national public health institutes (except in Lithuania where it is issued by the National Health Insurance Fund), whereas regional tenders are issued by regional public health institutes, regional governments, or regional tender organizations. In Belgium and France, most pharmacies purchase vaccines directly from the manufacturers, whereas vaccines are mostly purchased from wholesalers in Germany and Greece.

In some countries, multiple procurement systems are in place. In Spain, a national framework contract ("acuerdo marco") for influenza vaccine procurement is negotiated by the government, and regions can choose to procure vaccines through this contract or to issue their own regional tenders. In England and Wales, vaccines for children's programs (LAIV) are purchased centrally, vaccines for all other eligible groups are procured directly by GPs, and to a lesser extent by pharmacies [9,14]. Whilst in Belgium the bulk of vaccine procurement takes place through pharmacies, a regional tender is organized by the Flemish Community for healthcare institution residents, and multiple tenders for vaccination of occupational risk groups are in place.

Procurement systems can change over time. In Scotland, influenza vaccines were directly purchased by pharmacies until the 2015/16 season, and centralized procurement took place as of the 2016/17 season through national public tenders [15]. In Germany, public tenders were used for the procurement of vaccines up to the 2017–2018 season, before switching to direct procurement for the 2018–2019 season.

National or regional public tenders can be issued annually or as multi-year tenders (Fig 2). Multi-year tenders consist of one or two-year contracts with additional optional years, up to a

| | | | Calendar time of the year prior to the start of vaccination campaigns | | | | | | | | | | | | | |
|---|---|---|---|---|---|---|---|---|---|---|---|---|---|---|---|---|
| Country | Contract duration | Pop. size (millions) | Oct | Nov | Dec | Jan | Feb | Mar* | Apr | May | Jun | Jul | Aug | Sep | Oct | Nov |
| Ireland | Annual | 4.8 | | ■ | | | | | | | | | | | 💉 | |
| Croatia | Multiyear (3) | 4.1 | | | ■ | | | | | | | | | | | 💉 |
| Netherlands | Multiyear (1+3) | 17.2 | | | ■ | ■ | | | | | | | | | 💉 | |
| Denmark | Multiyear (1+2) | 5.8 | | | | ■ | ■ | | | | | | | | | 💉 |
| Finland | Multiyear (1+3) | 5.5 | | | | ■ | ■ | | | | | | | | 💉 | |
| Lithuania | Annual | 2.8 | | | | | | ■ | | | | | | 💉 | | |
| Scotland | Annual | 5.4 | | | | | | ■ | ■ | | | | | | 💉 | |
| Norway | Multiyear (2+2) | 5.3 | | | | | | | ■ | | | | | | 💉 | |
| Slovenia | Annual | 2.1 | | | | | | | ■ | | | | | | | 💉 |
| Sweden | Multiyear | 10.1 | ■ | ■ | ■ | | | | | | | | | | | 💉 |
| Italy | Annual** | 60.5 | | | | | | | | | ■ | ■ | ■ | ■ | 💉 | |
| Spain | Annual* | 46.7 | | | | | | | | | ■ | ■ | ■ | ■ | 💉 | |
| England | Annual | 56.0 | ■ | ■ | ■ | ■ | ■ | | | | | | | | 💉 | |
| Wales | Annual | 3.1 | | ■ | ■ | ■ | ■ | | | | | | | | 💉 | |
| France | Annual | 66.9 | | | ■ | | | | | | | | | | 💉 | |
| Belgium | Annual | 11.4 | | | ■ | ■ | ■ | | | | | | | | 💉 | |
| Germany | Annual | 82.8 | | | ■ | ■ | ■ | ■ | | | | | | | 💉 | |
| Greece | Annual | 10.7 | | | | | | | | | | | | ■ | 💉 | |

■ Expected award of national public tender

■ Expected awared of regional public tenders

■ Expected placement of majority of bookings by GPs

□ Expected placement of majority of bookings by pharmacies

💉 Start of vaccination campaigns in 2019

**Fig 2. Expected timeframe of tender award or placement of majority of direct purchase bookings [16,17].** (x+y): x years + y optional years (optional for the buyer). *A small number of regions have multi-year tenders. Source population size: EUROSTAT and Office of National Statistics [16,17].

maximum contract duration of four years. Procurement based on direct purchase generally concern single season contracts.

## Timing of influenza vaccine procurement

The earliest time at which vaccine brand availability can be known is the time when a public tender is awarded or when pre-bookings for direct purchase are placed (Fig 2). However, expected vaccine type availability may be known at the time when a tender is issued, or may be inferred from type-specific vaccine recommendations in settings where these exist.

National tenders are generally awarded relatively early, mostly between January and April prior to the influenza campaign, and the tender outcome is generally available in the public domain. In five countries, the contracts span several (optional) years, which, in the absence of major revisions to the national influenza vaccine recommendations, makes it possible to predict even further in advance which brands will be available in upcoming influenza vaccination campaigns.

Regional tenders in Sweden may be awarded up to 15 months prior to the influenza vaccination campaigns (and include multi-year contracts), but in Italy and Spain regional tenders are often awarded very close the start of the vaccination campaigns. Separately, there may exist a delay between the award of the tender and the publication of the tender outcome online.

In settings with direct purchase from the manufacturer, procurement information would exist once orders are placed. The timing when the majority of bookings are placed in settings with direct purchase varies widely across countries. Pre-bookings may be done early, for example at the end of the year prior to the season, such as in England or France; whereas in Greece, whilst pre-bookings may be done as of March, the majority of the procurement takes place much later, around September.

## Public availability of information on procured vaccines

With the exception of Sweden and Ireland, most tender outcomes for national and regional procurement systems are available online, such as on tender websites or public health institute websites (Tables 1 and 2). For Ireland, the tender outcome was obtained upon request from the public health institute. We were not able to systematically find tender outcomes for Sweden. Tender outcomes are often written in local languages and especially the regional tender outcomes are difficult and not always possible to find.

National procurement data can be informative of brand availability at clinic level. The brand available at clinic level is known when a single brand is purchased (such as in Croatia, Ireland, Lithuania and Slovenia), a single brand is purchased for use in a particular age group (e.g. QIVe in Finland in those <2y and ≥7y), or can be narrowed down to one of two vaccine brands (Denmark, Netherlands, and those ≥18y Norway). When available, regional procurement data may be informative of brand availability at clinic level. This depends on the number of vaccines purchased, the number of vaccine brands per type purchased and the availability of type-specific vaccine recommendations.

No readily accessible public and central source of vaccine brand information exists for settings in which vaccines are purchased by GPs or pharmacies. Whereas data from public sources such as the French reimbursement claims database Medic'am [26] and the English NHS Prescription Cost Analysis Data [27] can be useful to determine past vaccine brand use at country-level, these sources do not provide information prior to the vaccination campaigns. At country-level, information on vaccine types of brands may be available from other sources such as vaccine recommendations or news articles. Deducing vaccine brand availability at

**Table 1. Vaccines types and number of brands procured through national public tenders (2019–2020 season).**

| Country | Tender issuer | Number of vaccine brands by type at country level | Proportion per brand (as per tender) | Volume tendered (approximate) | Tender outcome publicly available, source | Can brand availability at clinic level be deduced? |
|---|---|---|---|---|---|---|
| **Croatia** | HZJZ | 1 TIV | NA–only 1 brand per type | 0.36 million | Yes, Oglasnik [18], TED [19] | Yes |
| **Denmark** | SSI | 2 QIVe | 40%/60% | 0.65–0.95 million | Yes, TED [19] | Partially, one or both of 2 brands is used at clinic level |
| **Finland** | THL | 1 QIVe, 1 LAIV* | NA–only 1 brand per type | QIV: 1.7 million | Yes, TED [19] | Yes, except for ages 2–6 years who may receive either QIV or LAIV |
| **Ireland** | HSE | 1 QIVe | NA–only 1 brand per type | 1 million | Obtained upon request from HSE | Yes |
| **Lithuania** | VLK | 1 QIVe | | 0.14 million | Yes, CVP IS [20] | Yes |
| **Netherlands** | RIVM | 2 QIVe | 40%/60% | 3.3 million | Yes, TenderNed [21], TED [19] | Partially, one or both of 2 brands is used at clinic level |
| **Norway** | FHI | 2 QIVe, 1 LAIV* | QIV: 40%/60% | QIV: 0.6–0.9 million | Yes, TED [19] | Partially, one of 2 QIV brands is used at clinic level (except for ages 2-17y who may receive either QIV or LAIV). Each region is assigned one QIVe brand, however which region is assigned which QIVe brand is not publicly known. |
| **Scotland** | NHS NSS | 1 aTIV, 1 QIVc, 2 QIVe, LAIV | Not reported | Not reported | Yes, Public Contracts Scotland [15] | No |
| **Slovenia** | NIJZ | 1 QIVe | NA–only 1 brand per type | 0.1 million | Yes, eNaročanje [22] | Yes |

*Purchase from manufacturer without tender procedure adjuvanted trivalent influenza vaccine (aTIV); CVP IS: Central Procurement Information System; FHI: Norwegian Institute of Public Health; HSE: Health Service Executive; HZJZ: Croatian Institute of Public Health; LAIV: Live attenuated quadrivalent influenza vaccine; NA: Not applicable; NHI: National health institute; NHS NSS: National Health Service National Services Scotland; NIJZ: National Institute of Public Health; QIVc: Cell-based quadrivalent influenza vaccine; QIVe: Egg-based quadrivalent influenza vaccine; RIVM: National Institute for Public Health and the Environment; SSI: Statens Serum Institut; TED: Tenders Electronic Daily; THL: National Institute for Health and Welfare; TIV: Conventional trivalent influenza vaccine; VLK: National health insurance fund.

clinic level is normally not possible, except when a single vaccine type for which a single vaccine brand is available is recommended for a specific risk group.

## Numbers of vaccine types and brands and historical variability

The number of vaccine types and brands available across countries in 2019–2020 is shown in Tables 1–3. The availability differed per country, but commonalities were observed per vaccine procurement mechanism at national, regional or clinic level. Alongside this information, whether the procurement information is informative on brand availability at the level of the clinic is shown. Vaccine types purchased used to be relatively stable across seasons, however the switch from trivalent to quadrivalent vaccines and the introduction of new vaccines such as QIVc and TIV-HD have caused changes in the past two seasons.

**National public tenders.** With one or two vaccine brands of a single vaccine type generally being procured through national tenders, type and brand diversity in these countries is low (Table 1). One exception is Scotland, where multiple vaccine types and brands are procured at national level. In Denmark and in the Netherlands two brands of QIVe (previously: TIV) are typically purchased and distributed across GP clinics. GP clinics in these two countries receive multiple deliveries of vaccines for the campaign and the brand may change between the deliveries, so either one or both of the vaccines may be used at each GP clinic, but this is not known ahead of time. In Norway two brands of QIVe are purchased and are

**Table 2. Trends in vaccine types and number of brands procured through regional public tenders (2019–2020 season).**

| Country | Regions | Tender issuer | Tender | Number of vaccine brands by type at region level | Number of vaccine brands at country level | Volume tendered | Tender outcome publicly available, source | Can brand availability at clinic level be deduced? |
|---|---|---|---|---|---|---|---|---|
| **Italy** | 21 | Regional govt/ tender organizations | Each region issues a tender | Generally 2 (1 egg-based QIVe and 1 aTIV) or 3 (1 QIVe, 1 aTIV, 1 QIVc) per region | 5 | Total: 11.5 million (range per region: 18,000 doses to 1,63 million doses) aTIV: 3,3 million (range per region: 0 to 490,000 doses) QIVe: 7,9 million (range per region: 5,000 to 1,5 million doses) QIVc: 290,000 (range per region: 0 to 230,000) | Generally yes, regional websites (in Italian) | Partially, if type-specific vaccination recommendations are followed, brand availability can often be deduced to a certain extent based on the population served |
| **Spain** | 17 | MoH, regional tender organizations | Independent procurement at region level National framework agreement** | Varies, ≥1 per region In framework agreement: 1 TIV, 1 aTIV, 2 QIVe [23] | ≥6 | Varies Through framework agreement: 3 million [24] | Generally yes, regional websites; Boletín Oficial del Estado [25] (in Spanish), TED [19] | Partially, if regional type-specific recommendations are in place and followed, brand availability may be deduced to a certain extent based on the population served |
| **Sweden** | 9* | Regional govt/ tender organizations | Each (cluster of) regions issued a tender | Generally 1 QIVe per region. | 3 | Unknown | Varies, limited information for some regions on TED [19] | Yes if tender outcome is available. No if tender outcomes is not available. |

*Sweden has 21 regions, but clusters of regions issue tenders.

**In 2019–2020, the following purchased influenza vaccines through the framework agreement: 9 autonomous communities (Aragón, Baleares, Cantabria, Castilla y León, Extremadura, Madrid, Murcia, Navarra and La Rioja) and two cities (Ceuta and Melilla) [24].

aTIV: Adjuvanted trivalent influenza vaccine; govt: Government; MoH: Ministry of Health; QIVc: Cell-based quadrivalent vaccine; QIVe: Egg-based quadrivalent influenza vaccine; LAIV: Live attenuated influenza vaccine; TIV: Conventional trivalent influenza vaccine.

**Table 3. Vaccine types and number of brands available in countries with vaccine procurement by pharmacies or GPs (2019–2020 season).**

| Country | Purchaser | Number of vaccine brands by type at country-level | Public availability information on bookings | Can brand availability at clinic level be deduced? |
|---|---|---|---|---|
| **Belgium** | Pharmacies | 3 QIVe [28] | No | No |
| **France** | Pharmacies | 2 QIVe [29] | No | No |
| **Germany** | Pharmacies | 3 QIVe, QIVc [30] | No | No |
| **Greece** | Pharmacies | QIV, number unknown | No | No |
| **England** | GPs (mostly) | ≥3 QIVe, QIVc, aTIV, 1 LAIV, 1 TIV-HD (TIV-HD is not reimbursed) [31] | No | No |
| **Wales** | GPs (mostly) | 2 QIVe, 1 QIVc, 1 aTIV, 1 LAIV, 1 TIV-HD [14] | No | No |

aTIV: Adjuvanted trivalent influenza vaccine; GPs: General practitioners; HD: High dose; LAIV: Live-attenuated quadrivalent influenza vaccine; QIVc: Cell-based quadrivalent vaccine; QIVe: Egg-based quadrivalent influenza vaccine; TIV: Conventional trivalent influenza vaccine; TIV-HD: High-dose trivalent vaccine.

allocated to separate regions. A single brand of QIVe is purchased in Finland, Ireland and Slovenia. In addition, LAIV vaccine (of which there is currently only one brand) is procured for use in children in Norway and Finland.

**Regional public tenders.** Vaccine type and brand diversity at the country level in Italy and Spain, where regional public tenders are in place, is higher than for countries with a national public tender (Table 2), however brand diversity at regional level varies. In Italy, generally two to three brands were procured in each region for the 2019–2020 campaign, resulting in a total of five brands at the country-level. In Spain, one or multiple vaccine types are procured in each region, resulting in at least 6 brands used across the country.

In Sweden, one of two brands of QIVe is procured in each region. However procurement data is not readily accessible in the public domain for all regions.

**Direct purchase by pharmacies and GPs.** Whilst vaccine type diversity in countries with direct purchase is similar to or higher than diversity in countries with national public tenders, brand diversity is typically higher (Table 3). In 2019–2020, in countries with direct purchase by pharmacies, vaccine type and brand diversity was highest in Germany with two types and four brands (three brands of 3 QIVe, one brand of QIVc), compared to one type and two or three brands (of QIVe) in France and Belgium, respectively. In England and Wales, where most vaccines are directly purchased by GPs, five vaccine types and multiple vaccine brands (three brands of QIVe in England and two brands of QIVc in Wales, and one each of TIV, aTIV and LAIV) are available in 2019–2020.

## Discussion

Four main systems for influenza vaccine procurement for use in national campaigns were identified across 18 European countries. Timely and publicly accessible information on brand availability in an upcoming influenza season is limited, especially for countries without national or multiyear tenders.

Procurement systems which are characterized as national level procurement with multiyear tenders and limited brand diversity, such as in Croatia, the Netherlands, Denmark, Finland and Norway, best allow to prospectively determine vaccine availability. This will also ensure adequate volume of a brand-specific vaccine coverage provided overall vaccine coverage is high. However, the trade-offs are that only few such situations exists and in those systems few brands may be captured in a given study.

Half of the surveyed countries have direct purchase mechanisms or regional tenders in place. While these systems often have a wider variety of vaccine brands with varying proportions per brand, information on type or brand availability comes late, if at all. Due the time required to prepare research sites for participation in VE research, late knowledge on vaccine brand availability is a challenge when aiming to engages specific sites based on their vaccine brands available in a VE study. In addition, though more brands may be captured, the volume per brand and consequently brand-specific vaccine coverage may be limited.

Wilsdon et al. studied the relationship between procurement method, market dynamics and public health in several European countries [32], and also found that tenders are associated with the availability of fewer different vaccines. Furthermore they describe the impact of procuring vaccines through tenders on vaccine price, supply sustainability, value-added services and incentivizing innovation, and suggest sustainable approaches to vaccine procurement.

Knowledge on brands that will be available in upcoming influenza seasons could inform site-selection for prospective observational brand-specific VE studies in DRIVE, to attain sufficient sample size per brand. However, the usefulness of procurement data to prospectively select locations for observational studies assessing brand-specific VE is limited.

## Strengths and limitations

It is acknowledged that the manufacturers hold information on the brand availability that may be of value for the implementation of the brand-specific VE studies. Manufacturer marketing data is however considered sensitive information and thus subject to provisions of EU competition law if not otherwise publicly available. The IFPMA's Influenza Vaccine Supply International Task Force has previously successfully used data from manufacturers. Through use of the global distribution of influenza vaccine doses, collected from manufacturers through a mechanism ensuring confidentiality, the IFPMA approximated influenza vaccination coverage [33]. However, to maintain compliance with the competition laws data was collected retrospectively, not type- or brand-specific and it was aggregated by WHO region (not by country). As such, this data could not be utilized to determine brand availability in upcoming influenza seasons for the purpose of supporting the prospective implementation of VE surveillance.

The expert interview was completed for 16 countries in Europe and online information was found for another two; several attempts were made to contact additional countries but no response was received. The interview was restricted to the national influenza vaccination programs, and only describes the main procurement system that is in place in each country. Whilst the data was updated to reflect the most recent 2019–2020 season, the vaccine types and brand procured are subject to change every season. In addition, the procurement system itself may also be subject to change, as illustrated by the recent change in Germany.

Although procurement data likely gives a solid basis to determine vaccine brand availability, information on expected vaccine volume obtained prior to the season may not reflect actual use, as additional orders may be booked and excess vaccine may be returned to the manufacturers at the end of the season.

Finally, we acknowledge that the current work is not exhaustive on all aspects of the procurement policy. Nevertheless, it addresses a gap in the literature, and can serve as a stepping stone for further research.

## Conclusion

Four approaches to the procurement of influenza vaccines have been identified in European countries. Countries for which vaccine procurement data for an upcoming influenza season is available and timely have a low diversity of vaccine types and brands (generally only one or two). Combined with the small number of countries for which this applies, the usefulness of procurement data to prospectively select locations for observational studies assessing brand-specific VE is limited.

## Supporting information

**S1 Appendix. Interview guidance.**
(DOCX)

**S2 Appendix.**
(PDF)

## Acknowledgments

We would like to acknowledge the following experts for sharing their expertise on procurement of influenza vaccines: Hélène Bricout (Sanofi Pasteur), Maura Cambiaggi (Seqirus), Javier Diez-Domingo (FISABIO), Frank Eberlein (Seqirus), Staša Javornik (NIJZ), Knut JØnsrud (FHI), Bernard Kaić (HZJZ), Paul Kenny (GlaxoSmithKline), Sjirk Kok (RIVM),

Toni Lloret (Seqirus), Simon de Lusignan (University of Surrey, University of Oxford), Birgit Neale (SSI), Hanna Nohynek (THL), Alice Pauc (Sanofi Pasteur), Austė Marija Strakšaitė (VLK); and Philippe Beutels (University of Antwerp) for reviewing the manuscript.

## Author Contributions

**Conceptualization:** Mendel Haag.

**Formal analysis:** Anke L. Stuurman.

**Investigation:** Anke L. Stuurman, Mendel Haag.

**Methodology:** Anke L. Stuurman, Caterina Rizzo, Mendel Haag.

**Writing – original draft:** Anke L. Stuurman.

**Writing – review & editing:** Caterina Rizzo, Mendel Haag.

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
