## [Decision Letter · Decision Letter 0]

9 Dec 2020

PONE-D-20-17950

Investigating the procurement system for understanding seasonal influenza vaccine brand availability in Europe

PLOS ONE

Dear Dr. Stuurman,

Thank you for submitting your manuscript to PLOS ONE. After careful consideration, we feel that it has merit but does not fully meet PLOS ONE’s publication criteria as it currently stands. Therefore, we invite you to submit a revised version of the manuscript that addresses the all points raised below during the review process.

We look forward to receiving your revised manuscript.

Kind regards,

Ray Borrow, Ph.D., FRCPath

Academic Editor

PLOS ONE

Journal Requirements:

"MH is an employee of Seqirus."

We note that one or more of the authors are employed by a commercial company: Seqirus, P95 Epidemiology and Pharmacovigilance.

2.1. Please provide an amended Funding Statement declaring this commercial affiliation, as well as a statement regarding the Role of Funders in your study. If the funding organization did not play a role in the study design, data collection and analysis, decision to publish, or preparation of the manuscript and only provided financial support in the form of authors' salaries and/or research materials, please review your statements relating to the author contributions, and ensure you have specifically and accurately indicated the role(s) that these authors had in your study. You can update author roles in the Author Contributions section of the online submission form.

2.2. Please also provide an updated Competing Interests Statement declaring this commercial affiliation along with any other relevant declarations relating to employment, consultancy, patents, products in development, or marketed products, etc.  

4. We note that Figure 1 in your submission contain map images which may be copyrighted. All PLOS content is published under the Creative Commons Attribution License (CC BY 4.0), which means that the manuscript, images, and Supporting Information files will be freely available online, and any third party is permitted to access, download, copy, distribute, and use these materials in any way, even commercially, with proper attribution. For these reasons, we cannot publish previously copyrighted maps or satellite images created using proprietary data, such as Google software (Google Maps, Street View, and Earth). For more information, see our copyright guidelines: http://journals.plos.org/plosone/s/licenses-and-copyright.

4.1.    You may seek permission from the original copyright holder of Figure 1 to publish the content specifically under the CC BY 4.0 license. 

4.2.    If you are unable to obtain permission from the original copyright holder to publish these figures under the CC BY 4.0 license or if the copyright holder’s requirements are incompatible with the CC BY 4.0 license, please either i) remove the figure or ii) supply a replacement figure that complies with the CC BY 4.0 license. Please check copyright information on all replacement figures and update the figure caption with source information. If applicable, please specify in the figure caption text when a figure is similar but not identical to the original image and is therefore for illustrative purposes only.

Reviewers' comments:

Reviewer's Responses to Questions

**Comments to the Author**

1. Is the manuscript technically sound, and do the data support the conclusions?

Reviewer #1: Partly

Reviewer #2: Partly

2. Has the statistical analysis been performed appropriately and rigorously? 

Reviewer #1: N/A

Reviewer #2: Yes

3. Have the authors made all data underlying the findings in their manuscript fully available?

Reviewer #1: Yes

Reviewer #2: Yes

4. Is the manuscript presented in an intelligible fashion and written in standard English?

Reviewer #1: Yes

Reviewer #2: Yes

5. Review Comments to the Author

Reviewer #1: This is a very important topic but I did not feel the paper addressed the core question of whether European countries are getting access to safe and effective influenza vaccines at competitive prices. I am not aware of DRIVE and how this assesses Vaccine Effectiveness (VE) in Europe. Presumably this can be only done "after the fact" i.e. manufacturers will produce their vaccines based on WHO recommendations of anticipated circulating strains and subsequent vaccines may be variably matched resulting in variable VE. As a reader, I am more interested to know how many influenza vaccines are potentially available globally, how many are assessed by EMA, how many are licensed by EMA, and what are the differential prices. Then I would like to know which of the 4 procurement mechanisms identified can obtain vaccines at the most competitive prices. Obviously we know that industry often likes to keep their pricing opaque for commercial reasons and we know that likely different countries in Europe many be paying different prices for the same vaccines. This paper highlights 4 main types of procurement mechanisms - from national tender systems choosing only one Product (presumably what is considered the cheapest safe and effective vaccine offered during the tender process). Whereas countries where GPs and Pharmacies buy vaccines from manufacturers have more brands (presumably all safe and effective but likely all at higher prices than the single product purchased through national tender). To improve this paper, I think there needs more explanation of the EMA system of licensing, more details of the who the 10 national experts are and who the 5 companies are that participated in the interviews. Given that one of the authors works for a vaccine manufacturer, and that 5 companies contributed data, and that the conclusion seems to be national tender systems have access to less brands, I am concerned that paper gives the impression that "access to more brands is better". The fact that there is no mention of pricing and which of these systems procure vaccines at lower pricing is of concern, since this should be the core interest of health policy makers and regulators i.e. as a policy maker I would be interested to know whether National Tender systems or regional tender systems should be the norm for Europe if it can be shown that those countries using using these systems can purchase lower priced vaccines. Should Europe develop a pooled procurement system similar to the PAHO Revolving Fund? This could be an interesting addition to the discussion.

Reviewer #2: Comment on Overall Paper:

This manuscript is relatively average on describing vaccines procurement across European countries. My suggestion is rather than using telephone interview, it is better to widen and support the writing with each countries' profiles on vaccines procurement. It is suggested checklist/tool/assessment used are better to be spell out in this paper. Extensive descriptions on vaccines procurement can be explained, for example the flow chart on procurement process, stakeholders involve & country's procurement policy. After all this manuscript is moderately written, can be enhanced by elaborating on aspects mentioned.

6. PLOS authors have the option to publish the peer review history of their article (what does this mean?). If published, this will include your full peer review and any attached files.

Reviewer #1: No

Reviewer #2: No

---

## [Author Response · Author response to Decision Letter 0]

27 Jan 2021

Below please find the edits we have made to comply with the journal requirements: 

1. PLOS One style requirements: The following changes have been made: the referencing style has been changed to Vancouver, the header level font has been changed to 16, the corresponding author has been indicated with an asterisk, and the figure file names have been renamed to meet PLOS One’s requirements.

2. Competing Interests Statement: “MH is an employee of Seqirus. AS is an employee of P95 Epidemiology and Pharmacovigilance. This does not alter our adherence to PLOS ONE policies on sharing data and materials.”

Funding Statement: “The funder provided support in the form of salaries for authors AS, CR, MH, but did not have any additional role in the study design, data collection and analysis, decision to publish, or preparation of the manuscript. The specific roles of these authors are articulated in the ‘author contributions’ section.”

3. The ORCID iD for the corresponding author is 0000-0002-4157-1315 

4. Figure 1 in your submission contain map images which may be copyrighted: The map has been created using mapchart.net. A reference to the website has been added to the figure caption. On their website they state that “You are allowed to use, edit and modify any map created with mapchart.net for private or commercial use/publication freely by adding any kind of reference to mapchart.net in your work.” (https://mapchart.net/feedback.html).

Please also see the 'Response to Reviewers' file for a point-by-point response to the reviewers’ comments.

---

## [Decision Letter · Decision Letter 1]

9 Mar 2021

Investigating the procurement system for understanding seasonal influenza vaccine brand availability in Europe

PONE-D-20-17950R1

Dear Dr. Stuurman,

We’re pleased to inform you that your manuscript has been judged scientifically suitable for publication and will be formally accepted for publication once it meets all outstanding technical requirements.

Kind regards,

Ray Borrow, Ph.D., FRCPath

Academic Editor

PLOS ONE

Additional Editor Comments (optional):

Reviewers' comments:

Reviewer's Responses to Questions

**Comments to the Author**

1. If the authors have adequately addressed your comments raised in a previous round of review and you feel that this manuscript is now acceptable for publication, you may indicate that here to bypass the “Comments to the Author” section, enter your conflict of interest statement in the “Confidential to Editor” section, and submit your "Accept" recommendation.

Reviewer #1: All comments have been addressed

2. Is the manuscript technically sound, and do the data support the conclusions?

Reviewer #1: Yes

3. Has the statistical analysis been performed appropriately and rigorously? 

Reviewer #1: N/A

4. Have the authors made all data underlying the findings in their manuscript fully available?

Reviewer #1: Yes

5. Is the manuscript presented in an intelligible fashion and written in standard English?

Reviewer #1: Yes

6. Review Comments to the Author

Reviewer #1: This is a timely topic given the complexities of regulations and requirements for vaccine licensing and access in Europe (and elsewhere). Although the revisions have not addressed all my previous suggestions, I accept that the paper contributes to an understanding of this topic and limitations of the scope of the paper have been added.

7. PLOS authors have the option to publish the peer review history of their article (what does this mean?). If published, this will include your full peer review and any attached files.

Reviewer #1: No

---

## [Editor Report · Acceptance letter]

31 Mar 2021

PONE-D-20-17950R1 

Investigating the procurement system for understanding seasonal influenza vaccine brand availability in Europe 

Dear Dr. Stuurman:

I'm pleased to inform you that your manuscript has been deemed suitable for publication in PLOS ONE. Congratulations! Your manuscript is now with our production department. 

Kind regards, 

on behalf of

Prof. Ray Borrow 

Academic Editor

PLOS ONE